# Effect of Percutaneous Biliary Drainage on Enzyme Activity of Serum Matrix Metalloproteinase-9 in Patients with Malignant Hilar Obstructive Hyperbilirubinemia

**DOI:** 10.3390/medicina59020336

**Published:** 2023-02-10

**Authors:** Aleksandar Filipović, Dragan Mašulović, Kristina Gopčević, Danijel Galun, Aleksa Igić, Dušan Bulatović, Miloš Zakošek, Tamara Filipović

**Affiliations:** 1Faculty of Medicine, University of Belgrade, 11000 Belgrade, Serbia; 2Center for Radiology, University Clinical Centre of Serbia, 11000 Belgrade, Serbia; 3Institute for Chemistry in Medicine, 11000 Belgrade, Serbia; 4HPB Unit, Clinic for Digestive Surgery, University Clinical Centre of Serbia, 11000 Belgrade, Serbia; 5Institute for Rehabilitation, 11000 Belgrade, Serbia

**Keywords:** biliary drainage, matrix metalloproteinase, malignant hilar obstruction

## Abstract

*Background and Objectives*. Cholestasis activates complex mechanisms of liver injury and as a result has an increased production of matrix metalloproteinases (MMP). Depending on the stage of liver disease, different matrix metalloproteinases expressions have been detected and could serve as indirect biomarkers as well as therapeutic targets. MMP-9 proteolytic activity has a proven role in both liver regeneration and neoplastic cell invasion in various malignancies. The purpose of this prospective cohort study was to evaluate the effect of external biliary drainage on enzyme activity of MMP-9 in the serum of patients with malignant hilar biliary obstruction. *Materials and Methods*. Between November 2020 and April 2021, 45 patients with malignant hilar biliary obstruction underwent percutaneous biliary drainage following determination of serum MMP-9 enzyme activity (before treatment and 4 weeks after the treatment) by gelatin zymography. *Results*. MMP-9 values decreased statistically significantly 4 weeks after percutaneous biliary drainage (*p* = 0.028) as well as the value of total bilirubin (*p* < 0.001), values of direct bilirubin (*p* < 0.001), aspartate aminotransferase (AST) (*p* < 0.001), alanine transaminase (ALT) (*p* < 0.001), and gamma-glutamyl transferase (GGT) (*p* < 0.001). *Conclusions*. In patients with malignant hilar biliary obstruction treated by external percutaneous biliary drainage for cholestasis resolution, a significant reduction in MMP-9 serum values was noted 4 weeks after the treatment.

## 1. Introduction

The most frequent causes of malignant bile duct obstruction are cholangiocarcinoma, gallbladder cancer, pancreatic cancer, and metastatic disease. When a tumor within or adjacent to a bile duct significantly reduces the normal bile flow, cholestasis appears [1]. In case of malignant bile duct obstruction, the first step of treatment should be a biliary system decompression to lower the serum bilirubin, regenerate the liver, and allow for further therapy.

Low bile duct obstruction (below the insertion of the cystic duct) should be treated by endoscopy approach, whilst percutaneous approach is preferred in high bile duct obstruction (above the insertion of the cystic duct). In cases with low bile duct obstruction when endoscopic approach was not possible or not successful, a percutaneous approach could be considered [2].

Impairment of bile flow leads to the accumulation of bile acid and other metabolites in the liver and systemic circulation and results in liver injury [3,4,5]. Studies in the past had a main focus on the role of bile acids in cholestasis although new data suggest more complex mechanisms of liver injury [6]. Understanding of those mechanisms has a substantial role in the treatment of cholestatic liver injury. Therefore, experimental models that mimic various aspects of mechanisms during cholestasis that lead to hepatic inflammation and fibrosis have been generated [7]. High concentrations of bile acids are proven to induce cell death and lead to further inflammatory liver injury [8]. Accumulation of bile acids in hepatocytes leads to a sustained chemotactic signal and predisposes these hepatocytes towards a neutrophilic attack [6]. Infiltration of immune cells results in an increased production of matrix metalloproteinases (MMPs) [9]. MMPs are involved in liver regeneration due to their effect on remodeling of extracellular matrix (ECM) [10,11,12]. It has been proven that matrix metalloproteinase-9 (MMP-9) promotes liver regeneration [13,14].

MMP-9 has a proven role in different hepatic disease such as the following: liver fibrogenesis; alcoholic liver cirrhosis (stage C) and hepatocellular carcinoma (HCC); promoting tumor invasion and metastasis; contributing to loss of blood–brain barrier integrity and edema during acute liver failure; accelerating liver regeneration and improving early-stage brain injuries in acute liver failure/fulminant hepatic failure [15]. Depending on the stage of liver disease, different MMP expression patterns have been detected and therefore could be used as indirect biomarkers in liver diseases [15].

Based on a previously established role of MMP-9 in liver regeneration [10,11,12], we hypothesized that the enzymatic activity of MMP-9 is in correlation with cholestasis resolution after percutaneous biliary drainage.

There are no published data on the enzyme activity of serum MMP-9 in patients with malignant hilar obstructive hyperbilirubinemia after percutaneous biliary drainage (PBD).

The purpose of the study was to determine the use of MMP-9 as a potential biomarker among patients with hilar malignancies causing obstructive hyperbilirubinemia for earlier identification of change during treatment.

## 2. Materials and Methods

### 2.1. Study Design

The study was designed as a single-center-based prospective cohort study.

The study protocol was reviewed and approved by Ethics Committee of University Clinical Center of Serbia (protocol number 717/11, approval date 26 November 2020) and approved by ClinicalTrials.gov (NCT number: NCT04801914). During the research, all the appropriate measures were taken to protect the privacy and anonymity of the patients’ data. The research was performed in compliance with the relevant laws and institutional as well as international guidelines.

### 2.2. Participants

All participants were fully informed about the study design. After obtaining a written informed consent, patients were included in the study.

The inclusion criteria: malignant hilar biliary obstruction proven by imaging and laboratory data; ECOG performance status 0–1.

The exclusion criteria: cardiac and renal insufficiency, terminal phase of the disease, and ECOG performance status 2–4.

In order to minimize selection bias, all consecutive cases that met the study criteria were included.

Between November 2020 and April 2021, 49 patients with malignant hilar biliary obstruction underwent PBD, and the preprocedural blood samples were taken from 48 patients (1 declined to participate), but control samples 4 weeks after treatment were taken from only 45 patients. Three patients were lost to follow up due to the COVID-19 infection (Figure 1).

The clinical efficacy of percutaneous biliary drainage in study population was prospectively analyzed in relation to the clinical and laboratory parameters (before treatment and 4 weeks after the treatment).

### 2.3. Study Sample Size and Power

The sample size was calculated online (https://statulator.com/SampleSize/ss2PM.html# (accessed on 20 July 2022) based on 80% power, a 5% significance level, and a “medium” effect size of 0.5 related to paired samples. The calculator recommended a sample size of 34, and due to possible drop out, a larger sample was used.

### 2.4. Clinical and Laboratory Data Recording

Clinical and laboratory data were collected from all participants before intervention. Clinical data included age, gender, previous medical history, and cause of obstruction. Laboratory data included complete biochemical blood analysis. The cause and level of obstruction were determined based on preprocedural imaging and tumor markers.

At the follow-up examination, which was scheduled for all patients 4 weeks after biliary drainage, blood samples were taken for complete biochemical analyzes and samples for determination of serum MMP-9 activity.

### 2.5. Follow-Up

The first control examination for all participants was scheduled 24 h after PBD. For patients in whom the drainage catheter was adequately positioned and functioning at the follow-up examination and without complications in terms of bleeding and clinical deterioration, the next follow-up examination was scheduled in 4 weeks.

In a minor number of cases, at the first check-up (24 h after PBD), due to inadequate catheter position or insignificant hemobilia, correction of the catheter position was made, and in the next 7 days, several control examinations were carried out until the moment of appropriate catheter position verification or absence of hemobilia. For these patients, the next follow-up examination was scheduled 4 weeks from the appointment when biliary drainage was performed.

### 2.6. Intervention

#### 2.6.1. Percutaneous Transhepatic Biliary Drainage (PBD) Procedure 

PBD was performed under analgosedation with continuous monitoring of patients’ vital parameters. The initial puncture of the peripheral intrahepatic bile ducts was performed under ultrasound control, and the further course of the procedure was performed under continuous fluoroscopy (15 fps). The puncture was performed at an angle of less than 30 degrees [16] in order to reduce the possibility of adjacent blood vessel injury and at the same time ensure the smooth passage of the guide wire and the drainage catheter distally to one of the central intrahepatic bile ducts. We used a single-puncture technique and performed external biliary drainage by placing the 8.5 Fr drainage catheter with a 10 mm locking loop.

A procedure in which a drainage catheter was placed in the intrahepatic bile ducts and bile drainage was established into a drainage bag was considered a technically successful procedure. The main goal of the PBD procedure was to achieve optimal bile drainage volume in order to ensure efficient drainage and achieve normalization in serum bilirubin values within 4 weeks and complete absence of clinical signs and symptoms of cholestasis.

#### 2.6.2. Testing Procedures (Gelatin Zymography for the Assay of Enzyme Activity of Serum MMP-9)

Pre- and 4 weeks after percutaneous biliary drainage procedure (PBD), the blood samples were drawn from all patients for MMP-9 serum-level assessment. The blood samples were taken after 12 h overnight fast between 8.00–9.00 a.m. and collected in vacutainers, and serum was separated by centrifugation (3000 rpm), followed by storage at −24 °C for further examination by zymography [17,18,19].

The presence and enzymatic activity of MMP-9 in serum was determined by the zymography method on polyacrylamide gels in which gelatin was copolymerized [18,19].

Serum samples were subjected to protein separation by sodium dodecyl sulphate polyacrylamide gel electrophoresis (SDS-PAGE) in a gel containing co-polymerized protease substrate gelatin (0.1%). After separation in the gel, the proteases renatured and hydrolyzed the substrate in the gel. The presence and activity of MMP-9 in the sample was determined based on the decrease in staining with 0.05% (*w*/*v*) CBB G-250 (Coomassie Brilliant Blue R-250) digested gelatin at the sites of MMP activity [18,19].

The samples in which the protein content was previously determined and which were diluted appropriately with a 20% sucrose solution were pre-incubated at 37 °C for 40 min. The same volume of sample treatment buffer, which consists of the following components, was added to the samples activated in this way: 0.125 M Tris-HCl; pH 6.8; 20% glycerol; 10% SDS and 0.25% bromine phenol blue. The samples were then subjected to electrophoresis (20 µL per well) in the wells within the electrophoresis gel. After electrophoretic separation of proteins at a certain voltage and current (U = 150 V, I = 50 mA), gels were washed in solution Triton X-100 (2.5%, *v*/*v*) for 45 min to remove SDS from the gel, then washed twice in deionized water at +40 °C, and incubated for 24 h at 37 °C in enzyme assay buffer (50 mmol/L Tris-HCl; 0.2 M NaCl; 5 mmol/L CaCl_2_; 0.05% NaH3; pH 7.5) in a water bath (Lab-ThermKuhner Shaker, Kuhner, Switzerland) with stirring at 50 revolutions per minute. After incubation, the gels were stained with Coomassie Brilliant Blue R-250 (CBB G-250), with concentrations 0.05% in a mixture of methanol: acetic acid: water (2.5:1:6.5; *v*/*v*/*v*) and decolorized in 4% methanol with 8% acetic acid. For better sensitivity, the gels were further destained in a 1% solution of Triton X-100 (1–2 h). The appearance of bright zones on the dark background of the gel stained with CCB G-250 dye indicated the sites of gelatinase activity. The intensity of gelatinolytic activity was determined densitometrically using the ImageJ software program (ImageJ 1.48v software, National Institutes of Health, Bethesda, MD, USA). The obtained gelatinolytic activity of MMP-9 was determined in relation to the activity of standard MMP-9 enzyme (Sigma Aldrich, Merck, Darmstadt, Germany). Gels were photographed wet and stored at room temperature. Finally, the obtained gelatinolytic activities (arbitrary units) were quantified (ng/dL) as relative protease activities in relation to the relative protease activities of the standards for MMP-9 and recalculated according to the activities of the standards, which were applied in an amount of 5 ng/mL [20]. All reagents were from the Institute for Chemistry in Medicine, University of Belgrade, Faculty of Medicine (Sigma-Aldrich, Merck, Darmstadt, Germany).

### 2.7. Statistical Analysis

All statistical analyses were performed using the SPSS package program version 20.0 (IBM corporation, Chicago, USA). Results were presented as mean ± standard deviation. Student’s *t*-test for paired samples or Wilcoxon signed-ranks test was used to determine the difference between the basic and control values. Independent variables were selected based on clinical importance and plausible association with the MMP-9 level after procedure. The correlation between MMP-9 and bilirubin level, C-reactive protein (CRP), ALT, and AST at baseline was non-significant. Multivariate linear regression was used to estimate the association between MMP-9 values after procedure and clinical characteristics. All *p*-values < 0.05 (two-tailed) were considered significant.

## 3. Results

The study included 45 patients (24 men and 21 women). The average age of the population was 66.38 ± 11.28 (minimum 38 years, maximum 86 years).

In the study population, Bismuth type IV hilar cholangiocarcinoma (57.7%) was the most common cause of biliary obstruction (Table 1). There was no significant difference in MMP-9 serum values before and after PBD compared to the cause of malignant biliary obstruction (*p* = 0.199, *p* = 0.341, respectively).

Hilar metastases were present in seven patients (15.5%) and Bismuth type II cholangiocarcinoma in six patients (13.3%).

In all patients, PBD procedure was performed with technical success and without any significant complication as ruled out by clinical findings or follow-up imaging. Only a few patients had a minor and transient hemobilia. In the examined period, the value of total bilirubin (*p* < 0.001), values of direct bilirubin (*p* < 0.001), albumin concentration (*p* = 0.033), AST (*p* < 0.001), ALT (*p* < 0.001), ALP (*p* < 0.001), GGT (*p* < 0.001), alpha amylase (*p* < 0.001), monocytes (*p* < 0.001), and CRP were statistically significantly lower 4 weeks after percutaneous biliary drainage (control values) in comparison with the baseline values (Table 2). Urea (*p* < 0.001), total proteins (*p* < 0.001), sodium (*p* < 0.001), potassium (*p* = 0.013), chlorides (*p* < 0.001), calcium (*p* < 0.001), phosphorus (*p* < 0.001), magnesium, RBC 10~12 (*p* = 0.002), HGB g/L (*p* < 0.001), HCT L/L (*p* < 0.001), and PLT 10~9 (*p* = 0.035) were statistically significantly higher in the control compared to the basic values (Table 2).

MMP-9 values decreased statistically significantly during the monitoring period (baseline values: 595.5 ± 193.78 vs. 4 week after PBD: 371.2 ± 219.16, *p* = 0.028) (Figure 2).

Linear regression showed that presence of metastasis in the liver is significantly associated with MMP-9 values after procedures (Beta = −0.911, *p* = 0.049) when adjusted for baseline MMP-9 values, presence of lymphadenopathy, CRP, and glucose (Table 3).

## 4. Discussion

The extracellular matrix (ECM) forms a dynamic environment that undergoes continuous remodeling during development, differentiation, and wound-healing processes to maintain homeostasis and prevent disease onset and progression [21,22]. In all these processes, MMPs have a substantial role [23]. MMPs or matrixins are endo-proteinases that degrade ECM components, regulate ECM integrity and composition, and play an important role in ECM-mediated signaling [22].

Matrix metalloproteinases (MMPs) are a family of zinc- and calcium-dependent endopeptidases that are produced by epithelial cells, endothelial cells, fibroblasts, and inflammatory cells (monocytes, macrophages, and neutrophils) and mostly secreted in the pro-enzyme form and activated in the extracellular space [24]. Matrixins have a proven role in regulation of proliferation, differentiation, migration, adhesion, and apoptosis [22].

MMPs regulate cell behavior by cleaving cell-surface molecules and pericellular nonmatrix proteins [25]. Dysregulation of MMP expression and/or activity could contribute to progression of liver diseases [15]. MMPs are involved at different stages of liver disease, resolution, and regeneration although the underlying mechanisms remain largely unknown [15]. Some studies have investigated MMPs as a potential therapeutic targets for the resolution of liver diseases [15].

Thus far, more than 26 human MMPases are known [26]. Based on the specificity of their substrate and according to their function, human MMPs can be divided into the following six groups: collagenases (MMP-1, -8, and -13); stromelysins (MMP-3, -10, -11, and -17); gelatinases (MMP-2 and -9); matrilysins (MMP-7 and -26); membrane-type MMPs (MMP-14, -15, -16, -17, -24, and -25); and others (MMP-12, -19, -20, -21, -22, -23, -27, and -28) [22].

The structural domain of metalloproteinases consists of three parts: the proenzyme domain, the catalytic domain, and the C-terminal domain, which is thought to define the specific substrate. Zinc (Zn) in the catalytic domain interacts with the conserved cysteine in domain I and thus maintains the proenzyme in an inactive form. Gelatinases (MMP-2 and MMP-9) also have an additional domain with three cascade junctions that repeat and resemble fibronectin type II; they cleave the catalytic region and interact with collagens and gelatins [27]. Under normal circumstances, MMP expression is minimal. Their expression is regulated by numerous factors such as growth factors, cytokines, integrins, chemical agents, drugs, cellular stress, ECM proteins, morphological changes, and intracellular signal transduction induced by the expression of certain MMP genes [26]. The balance between the activity of MMPases and their inhibitors is strictly controlled under physiological conditions and under the influence of various factors involved in ECM remodeling.

MMP-9 inhibition was shown to improve survival and liver function at the early stage [14] by inhibiting liver inflammation and damage [28,29]. Liver-selective MMP-9 inhibition has a proven effect in accelerating liver regeneration [13].

MMPs have been implicated in the initiation, progression, and resolution of different liver diseases [22]. MMP-9 expression was found to be highly increased during acute and fulminant liver failure due to the acute and severe impairment of liver functions [30]. As a consequence of malignant hilar biliary obstruction, acute liver failure might occur with increased MMP-9 expression and contribution to brain extravasation due to the loss of blood–brain barrier integrity. In this clinical scenario but unfortunately not at the late stage, MMP-9 inhibition could improve survival, liver function, and brain injuries [14]. MMPs expression patterns in different stages of liver diseases determine their potential role as biomarkers and a target for novel therapeutic drug discovery [15].

This study performed on a moderately sized and carefully defined group is the first one in the literature to reveal changes of MMP-9 serum activity after PBD in patients with malignant hilar obstructive hyperbilirubinemia. During this study, we investigated changes in MMP-9 activity before and after percutaneous biliary drainage performed for cholestasis treatment in order to achieve liver regeneration. Using gelatin zymography as a molecular technique, the statistically significant reduction in MMP-9 serum values was noted four weeks after percutaneous biliary drainage (Figure 2). This specific time frame of 4 weeks for making a control measurement of serum biochemistry and MMP-9 values was chosen as the period of time expected for cholestatic liver injury resolution. Even when bilirubin levels normalize, liver function recovers only between 4 and 6 weeks after PBD [31].

In comparison with preprocedural values (basic values), a significant decrease of MMP-9 values was noted 4 weeks after percutaneous biliary drainage (control values) as well as the value of total bilirubin (*p* < 0.001), values of direct bilirubin (*p* < 0.001), albumin concentration (*p =* 0.033), AST (*p* < 0.001), ALT (*p* < 0.001), ALP (*p* < 0.001), and GGT (*p* < 0.001), which could serve as indicators of cholestatic liver injury resolution. Results of the present study suggest that there is a correlation of MMP-9 enzyme activity with cholestatic liver injury resolution in patients with malignant hilar biliary obstruction. This finding could point out MMP-9′s potential role as a therapeutic target for inducing liver regeneration in malignant hilar biliary obstruction allowing further therapy.

Cholestatic liver disease could be associated with progressive intrahepatic biliary fibrosis [32,33,34]. In cholestatic fibrosis, portal fibroblasts have a proven contribution to an excessive ECM accumulation by secretion of collagen [35,36,37]. The regression of liver fibrosis includes restoration, elimination of active myofibroblasts, and extracellular matrix degradation [38]. The degradation of ECM is the most important step of liver fibrosis resolution. MMP-9 was found to be involved in ECM degradation with an effect in increasing matrix permeability-enhancing leukocyte infiltration and inflammation, leading to impaired liver function during ischemia reperfusion injury.

MMPs are dynamic in angiogenesis and extracellular matrix remodeling [39] and were found to be involved in the new capillary sprouts formation and neo-angiogenesis, resulting in invasion and metastasis [40]. Since MMP-9 is secreted by fibroblasts and endothelial cells with a high potential to degrade collagen IV as an important component of a basement membrane, it has a proven role in the formation of new capillary sprouts and neo-angiogenesis, resulting in invasion and metastasis [15,40]. iNOS-derived NO secreted by cancer cells modulates MMP-9 production, thus contributing to neo-angiogenesis, invasion, and metastasis [40]. Due to their versatile function, MMPs have been implicated in many different diseases including cancer and metastases [41]. Matrix metalloproteinases (MMPs) have the ability to degrade basement membranes and may thus play an important role in extracellular matrix turnover in liver fibrosis and carcinogenesis [42].

MMP-9 proteolytic activity is essential for the mechanisms of neoplastic cell invasion in various malignancies [43]. Since the greatest part of the study population had a cholangiocarcinoma, for the purpose of evaluating tumor response to therapy and outcome prediction, it was important to use the appropriate molecular marker. MMP-9 was already proven as an invasion and migration factor of cholangiocarcinoma cells [44]. Kuyvenhoven et al. yielded no significant differences in MMP values between patients with or without hepatocellular carcinoma [42]. In the study group, only one patient had a hepatocellular carcinoma, which was without significant impact on the final result of the current study.

It has been proven that molecular insights of MMP-9 provide a better understanding of the pathophysiology of tumor invasion and metastasis [43]. Clinical studies indicated that the expression of MMP-9 biomarker in epithelial and lymphatic neoplasia is both an independent prognostic marker characterized by poor overall survival as well as a strong prognostic tool for patients undergoing surgical or adjuvant therapy [43]. A certain group of investigators has already noted MMP-9 as a therapeutic target. A novel therapeutic approach by use of a certain antineoplastic agent with effects on MMP-9 activity suppression in cholangiocarcinoma cells was reported [45]. The streptochlorin-induced apoptosis effects were reported via inhibition of ROS production, MMP collapse, and the subsequent activation of caspase-3 [45].

MMP-9 is proven to be involved in ECM degradation, increasing matrix permeability, enhancing leukocyte infiltration and inflammation, and leading to impaired liver function during ischemia reperfusion injury (IRI). Therefore, MMP-9 inhibition could be an approach for the treatment of IRI liver injury [15].

Currently, the only therapeutic option for end-stage chronic liver diseases with an excessive accumulation of extracellular matrix ECM resulting in fibrosis is transplantation. Other therapeutic options for fibrotic liver damage are limited. However, ECM remodeling is a prerequisite for recovery from hepatic fibrosis [46]. Thus, pharmaceutic modulation of MMP activity might be a therapeutic strategy to promote resolution of liver fibrosis [47]. Additionally, liver-selective MMP-9 inhibition prevents MMP-9-mediated VEGF cleavage, which enhances recruitment and engraftment of sinusoidal endothelial cell progenitor cells, thereby accelerating liver regeneration [13]. One of the studies focused on gaining insights into the function of MMP-9 found MMP-9 to have highly increased expression during acute liver failure and fulminant liver failure, with contribution to brain extravasation and edema due to the loss of blood–brain integrity [28]. Thus, MMP-9 inhibition improved survival, liver functions, and brain injuries at the early stage [14].

### Limitations of the Study

The study was designed as a single-center study and performed on a moderately sized group of patients.

## 5. Conclusions 

In patients with malignant hilar biliary obstruction treated by external percutaneous biliary drainage for cholestasis resolution, a significant reduction in MMP-9 serum values was noted.

The findings of the current study could provide the basis for further investigation on MMP-9 as a novel diagnostic tool and therapeutic target for inducing liver regeneration in patients with malignant hilar biliary obstruction.

## Figures and Tables

**Figure 1 medicina-59-00336-f001:**
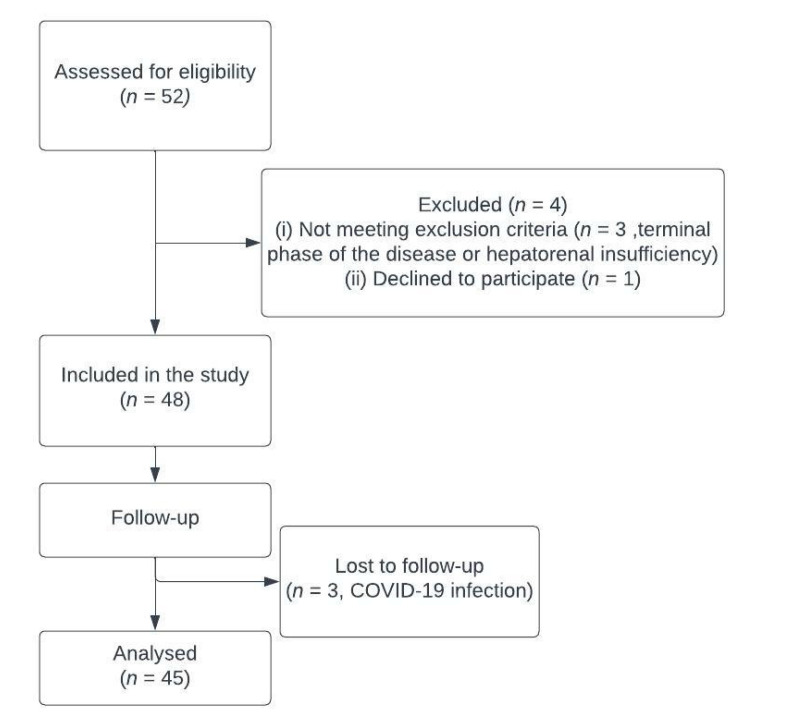
Participants flowchart.

**Figure 2 medicina-59-00336-f002:**
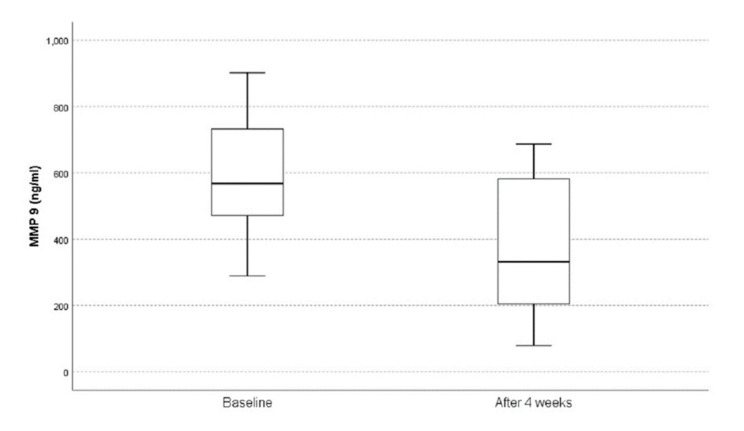
MMP-9 values in the follow-up period.

**Table 1 medicina-59-00336-t001:** Causes of malignant biliary obstruction.

	No. of Patients	%	MMP-9 before PBD	MMP-9 after PBD
Bismuth type IV hilar cholangiocarcinoma	26	57.7	648.7 ± 174.9	424.7 ± 141.9
Bismuth type II hilar cholangiocarcinoma	6	13.3	733.0 ± 239.0	470.0 ± 306.9
Hilar hepatocellular carcinoma	1	2.2	635.0	395.0
Hilar metastasis	7	15.5	491.5 ± 27.6	92.0 ± 18.4
Gallbladder carcinoma	5	11.1	798.5 ± 94.0	266.6 ± 188.5

MMP-9: matrix metalloproteinase-9; PBD: percutaneous biliary drainage.

**Table 2 medicina-59-00336-t002:** Biochemical parameters.

	Before PBD	4 Weeks after PBD	*p* ^1^
Glucose (mmol/L)	6.49 ± 1.76	6.22 ± 1.25	0.256
Urea (mmol/L)	7.44 ± 7.88	7.83 ± 7.17	<0.001
Creatinine (µmol/L)	114.02 ± 119.35	96.37 ± 78.99	0.183
Total bilirubin (umol/L)	384.02 ± 123.15	57.49 ± 39.29	<0.001
Direct bilirubin (umol/L)	229.31 ± 77.06	32.27 ± 24.92	<0.001
Total protein (g/L)	57.18 ± 8.51	60.91 ± 8.25	<0.001
Albumin (g/L)	33.69 ± 6.01	32.91 ± 5.18	0.033
Sodium (mmol/L)	135.87 ± 3.35	138.56 ± 4.08	<0.001
Potassium (mmol/L)	3.86 ± 0.59	3.99 ± 0.66	0.013
Chlorides (mmol/L)	102.98 ± 4.31	104.76 ± 5.66	<0.001
Calcium (mmol/L)	2.18 ± 0.16	2.22 ± 0.21	<0.001
Phosphorus	1.05 ± 0.19	1.11 ± 0.17	<0.001
Magnesium	0.69 ± 0.13	0.74 ± 0.13	<0.001
Aspartate transferase (U/L)	160.64 ± 102.3	86.18 ± 56.26	<0.001
Alanine transaminase (U/L)	196.53 ± 193.27	103.31 ± 87.14	<0.001
Alkaline phosphatase (U/L)	621.04 ± 411.1	240.96 ± 141.6	<0.001
Gamma-glutamyl transferase (U/L)	660.62 ± 480.14	255.49 ± 181.8	<0.001
Alpha amylase (U/L)	63.96 ± 63.81	41.47 ± 33.7	<0.001
White blood cells 10~9/L	8.61 ± 6.09	7.36 ± 2.01	0.865
Lymphocytes %	13.95 ± 7.27	14.36 ± 11.45	0.743
Monocytes %	8.01 ± 3.68	4.42 ± 2.24	<0.001
Neutrophils %	71.74 ± 14.82	72.28 ± 15.53	0.484
Red blood cells 10~12/L	3.73 ± 0.65	3.95 ± 0.68	0.002
Hemoglobin g/L	115.00 ± 19.25	124.09 ± 18.12	<0.001
Hematocrit L/L	0.34 ± 0.05	0.36 ± 0.06	<0.001
Platelet count 10~9/L	240.33 ± 79.78	248.56 ± 94.88	0.035
C-reactive protein (mg/L)	28.65 ± 41.94	23.78 ± 47.98	0.049

^1^*t*-test, Mann–Whitney test.

**Table 3 medicina-59-00336-t003:** Association MMP-9 values after PBD procedure with clinical parameters (multivariate linear regression analysis).

	Unstandardized Coefficients		Standardized Coefficients	Sig.
	(B)	Std. Error	Beta	
(Constant)	−516.295	427.747		0.294
MMP-9	1.051	0.397	0.929	0.057
Patients with metastatic lesions in liver	−386.709	138.206	−0.911	0.049
Lymphadenopathy	−146.625	141.718	−0.282	0.359
CRP	4.060	2.672	0.424	0.203
Glucose	75.154	37.548	0.585	0.116

Adjusted R^2^ = 0.498, B—standard error for unstandardized regression coefficient

## Data Availability

The data associated with the paper are not publicly available but are available from the corresponding author on reasonable request.

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
