# Peer review of "Effect of Percutaneous Biliary Drainage on Enzyme Activity of Serum Matrix Metalloproteinase-9 in Patients with Malignant Hilar Obstructive Hyperbilirubinemia"

_medicina, 2023, doi:10.3390/medicina59020336_

Round 1
Reviewer 1 Report
The authors are dealing with the important matter of the “Effect of Percutaneous Biliary Drainage on Enzyme Activity of Serum Matrix Metalloproteinase-9 in Patients with Malignant Hilar Obstructive Hyperbilirubinemia”. Based on the findings, the clinical application of this study is not enough currently because of small sample size and limitation in clinical use. I recommend revising below-mentioned point is addressed.
Q1: Can the authors explained how to choose the five factors for multivariate regression analysis? The discussion should include the explanations about why the liver metastasis is associated with MMP-9 values after procedures.
Q2: The authors can show more about the correlation between MMP-9 and bilirubin level (or CRP, ALT, AST…) at baseline. The association of clinical outcomes (survival, staging, response to treatment…) and MMP-9 level is also an interested issue for clinicians.
Q3: The clinical use of the study is limit because the clinicians can measure the bilirubin level to predict the treatment outcome. Can the authors try to discuss more about the clinical application in the future?
Q4: Line 208, “Table 1” should be corrected as “table 2”. The abbreviation in table 2 should be listed.
Author Response
Q1: Can the authors explaine how to choose the five factors for multivariate regression analysis? The discussion should include the explanations about why the liver metastasis is associated with MMP-9 values after procedures.
Reply: Independent variables were selected based on clinical importance and plausible association with the MMP 9 level after procedure.
In the discussion section following sentence is included: Since MMP-9 is secreted by fibroblasts and endothelial cells with a high potential to degrade collagen – IV as an important component of a basement membrane it has a proven role in the formation of new capillary sprouts and neo-angiogenesis resulting in invasion and metastasis. iNOS-derived NO secreted by cancer cells modulates MMP-9 production thus contributing to neo-angiogenesis , invasion and metastasis.
Q2: The authors can show more about the correlation between MMP-9 and bilirubin level (or CRP, ALT, AST…) at baseline. The association of clinical outcomes (survival, staging, response to treatment…) and MMP-9 level is also an interested issue for clinicians.
Reply: We thank the reviewer for this comment. However, the correlation between MMP-9 and bilirubin level, CRP, ALT, AST at baseline was non-significant, therefore is not presented here. We fully agree with the reviewer that further analysis of clinical outcomes and MMP-9 level association should follow the study that is presented in this paper. We intend to complete this analysis in the near future and to include more patients from longer observation period.
Q3: The clinical use of the study is limit because the clinicians can measure the bilirubin level to predict the treatment outcome. Can the authors try to discuss more about the clinical application in the future?
Reply: Although the underlying mechanisms are largely unknown MMPs have been implicated in the initiation, progression and resolution of different liver diseases. MMP-9 expresion was found to be highly increased during acute and fulminant liver failure due to the acute and severe impairment of liver functions. As a consequence of malignant hilar biliary obstruction acute liver failure might occur with increased MMP-9 expression and contribution to brain extravasation due to the loss of blood-brain barrier integrity. In this clinical scenario but unfortunately not at the late stage MMP-9 inhibition could improve survival, liver function and brain injuries. MMPs expression patterns in different stages of liver diseases determine their potential role as biomarkers and target for novel therapeutic drug discovery.
Q4: Line 208, “Table 1” should be corrected as “table 2”. The abbreviation in table 2 should be listed.
Reply: The correction is made in the manuscript.
Reviewer 2 Report
The manuscript describes the importance of MMP9 in malignant hilar biliary obstruction before treatment by external percutaneous 325 biliary drainage for cholestasis resolution. The study is well-conducted and presented.
1. A comparison between different liver pathologies and MMP9 serum levels can be added in Introduction.
2. the therapeutic and diagnostic potential as suggested by the authors can be elaborated with literature citations.
3. the limited sample size limitation is mentioned by the authors, but a systemic comparison (in a form of table) between different liver pathologies and MMP9 levels can help to strengthen the conclusion.
4. there are some minor grammatical and syntax errors.
Author Response
A comparison between different liver pathologies and MMP9 serum levels can be added in Introduction.
Reply: Based on the reviewer’s input the additional information regarding different liver pathologies and MMP9 serum levels has been added in the Introduction section. MMP-9 has a proven role in different hepatic disease as following: liver fibrogenesis; alchoholic liver cirrhosis (stage C) and HCC; promotes tumor invasion and metastasis; contributes to loss of blood-brain barrier integrity and edema during acute liver failure; accelerates liver regeneration and improves early stage brain injuries in acute liver failure/fulminant hepatic failure.
Table 1 is extended with additional data on MMP-9 serum values before and after PBD, depending on a liver pathology.
- the therapeutic and diagnostic potential as suggested by the authors can be elaborated with literature citations.
Reply: MMP-9 is proven to be involved in ECM degradation with increasing matrix permeability, enhancing leukocyte infiltration and inflammation leading to impaired liver function during ischemia reperfusion injury (IRI). Therefore MMP-9 inhibition could be an approach for the treatment of IRI liver injury.
Currently the only therapeutic option for end stage chronic liver diseases with an excessive accumulation of extracellular matrix ECM resulting in fibrosis is transplantation. Other therapeutic options for fibrotic liver damage are limited. However, ECM remodeling is a prerequisite for recovery from hepatic fibrosis. Thus pharmaceutic modulation of MMP activity might be a therapeutic strategy to promote resolution of liver fibrosis. Additionaly, liver selective MMP-9 inhibition prevents MMP-9- mediated VEGF cleavage that enhances recruitment and angraftment of sinusoidal endothelial cell progenitor cells, thereby accelerating liver regeneration. One of the studies focused on gaining insights into the function of MMP-9 have found MMP-9 highly increased expression during acute liver failure and fulminant liver failure with contribution to brain extravasation and edema due to the loss of blood-brain integrity. Thus MMP-9 inhibition improved survival , liver functions and brain injuries at the early stage.
- the limited sample size limitation is mentioned by the authors, but a systemic comparison (in a form of table) between different liver pathologies and MMP-R9 levels can help to strengthen the conclusion.
Reply: Table 1 is extended with additional data on MMP-9 serum values before and after PBD, depending on a liver pathology.
- there are some minor grammatical and syntax errors.
Reply: The necessary corrections are completed accordingly.
Round 2
Reviewer 1 Report
Thanks for the revised manuscript.
Can the authors explain the different presentation of MMP-9 in patients with type IV cholangiocarcinoma? Is the elevated MMP-9 correlated to recurrent cholangitis or obstructive jaunduce?
Author Response
Can the authors explain the different presentation of MMP-9 in patients with type IV cholangiocarcinoma? Is the elevated MMP-9 correlated to recurrent cholangitis or obstructive jaunduce?
Reply: We thank the reviewer for this comment. An error occurred when copying and entering data into Table 1. MMP 9 values after PBD in patients with Bismuth type IV cholangiocarcinoma are lower than before the procedure. The table is corrected and a new document with a certain number of words, in accordance with the magazine's propositions, uploaded.